# Medical diffusion on a budget: Textual Inversion for medical image generation

**Bram de Wilde**             CONTACT@BRAMDEWILDE.COM

**Anindo Saha**            ANINDYA.SHAHA@RADBOUDUMC.NL

**Maarten de Rooij**        MAARTEN.DEROOIJ@RADBOUDUMC.NL

**Henkjan Huisman**       HENKJAN.HUISMAN@RADBOUDUMC.NL

**Geert Litjens**             GEERT.LITJENS@RADBOUDUMC.NL

*Department of Medical Imaging, Radboud University Medical Center, Nijmegen, the Netherlands*

**Editors:** Accepted for publication at MIDL 2024

## Abstract

Diffusion models for text-to-image generation, known for their efficiency, accessibility, and quality, have gained popularity. While inference with these systems on consumer-grade GPUs is increasingly feasible, training from scratch requires large captioned datasets and significant computational resources. In medical image generation, the limited availability of large, publicly accessible datasets with text reports poses challenges due to legal and ethical concerns. This work shows that adapting pre-trained Stable Diffusion models to medical imaging modalities is achievable by training text embeddings using Textual Inversion. In this study, we experimented with small medical datasets (100 samples each from three modalities) and trained within hours to generate diagnostically accurate images, as judged by an expert radiologist. Experiments with Textual Inversion training and inference parameters reveal the necessity of larger embeddings and more examples in the medical domain. Classification experiments show an increase in diagnostic accuracy (AUC) for detecting prostate cancer on MRI, from 0.78 to 0.80. Further experiments demonstrate embedding flexibility through disease interpolation, combining pathologies, and inpainting for precise disease appearance control. The trained embeddings are compact (less than 1 MB), enabling easy data sharing with reduced privacy concerns.

**Keywords:** Diffusion models, Generative imaging, Low-resource, Prostate MRI, Chest X-ray, Histopathology

## 1. Introduction

Image generation has increasingly captured the attention of many researchers, spurring an impressive progression in text-to-image generation. In particular, diffusion models have gained enormous popularity through their ability to generate high-quality and diverse images, conditioned on a text prompt (Ho et al., 2020; Dhariwal and Nichol, 2021; Ramesh et al., 2021, 2022; Saharia et al., 2022). Among various text-to-image model implementations, Stable Diffusion has arguably generated the biggest impact in terms of users, owing to the fact that it is both released under a permissive license and operable using a single GPU (Rombach et al., 2022).

When applied to art or photorealistic images, generative models may exhibit some degree of error. On the other hand, the medical imaging field places a higher bar on generation quality (Yi et al., 2019; Skandarani et al., 2021). Images need to be not only anatomically

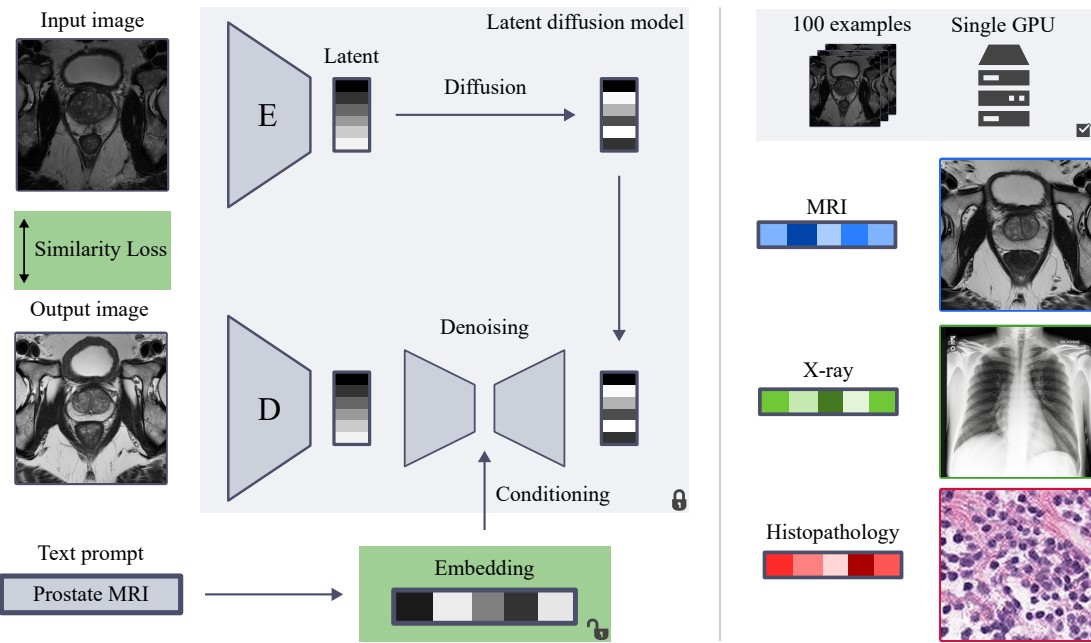

Figure 1: The Textual Inversion fine-tuning process for diffusion models trains a text conditioning embedding for a new token using a small set of images while keeping the rest of the architecture frozen. We show that this allows the adaption of latent diffusion models to a variety of medical imaging modalities, using only 100 examples and a single consumer-grade GPU.

correct but diagnostically correct as well. Training a model like Stable Diffusion for medical imaging requires a large, varied, and ideally public dataset of images with captions, similar to those used for training on natural images (Schuhmann et al., 2022). However, practical challenges, such as ethical and legal impediments to sharing medical data, particularly for unstructured radiology reports, complicate this endeavor (Scheibner et al., 2021; Bovenberg et al., 2020). For one of the few public datasets of this caliber that exists, MIMIC-CXR, Chambon et al. have demonstrated that it is possible to train a latent diffusion model capable of generating chest X-ray images with high fidelity and diversity through free text prompts (Johnson et al., 2019; Chambon et al., 2022a). They trained the system using up to 170,000 images on 64 A100 GPUs.

On top of data sharing issues, some modalities and pathologies are inherently scarce: certain types of scans can be expensive or experimental and some diseases are rare or tied to specific demographics. For these reasons, especially in the medical domain, it is essential to have computationally feasible methods that can fine-tune existing models towards a smaller set of a specific modality or disease. In this paper, we pick one such method, Textual Inversion, and rigorously explore its capacities for adapting Stable Diffusion to medical imaging, with all experiments performed on a single RTX2070 GPU (Gal et al., 2022). Code and trained embeddings are shared online.[1]

---

1. https://github.com/brambozz/medical-diffusion-on-a-budget

## 2. Related work

Several papers have applied diffusion to medical imaging, with a wide range of applications including anomaly detection, segmentation, registration, and modality transfer with image-to-image translation (Kazerouni et al., 2022). Specifically for medical image generation, several recent works have trained diffusion models for image generation. Pre-trained models are often trained on 2D RGB datasets, but many medical imaging modalities are 3D. Recently, studies such as Khader et al. (2023) and Pinaya et al. (2022) have trained diffusion models from scratch on 3D data or even on 4D data (Kim and Ye, 2022), and Han et al. (2023) use diffusion models conditioned on anatomical masks to generate labeled images for segmentation. Several other works studied text-to-image latent diffusion models for medical imaging (Chambon et al., 2022a; Akrout et al., 2023). Closest to our work is (Chambon et al., 2022b), where the authors explore various methods to adapt a pre-trained Stable Diffusion model to chest X-ray generation. They performed experiments with both Textual Inversion and fine-tuning the U-net component of Stable Diffusion, similar to (Ruiz et al., 2022). They find that Textual Inversion works, but fine-tuning the U-net is more effective, especially with more complex prompts. They fine-tune using 5 examples per class.

Our work builds on this by deeply exploring Textual Inversion by training with more examples and bigger embeddings. Additionally, we demonstrate the flexibility of the approach through example applications and by adapting to multiple and more complex modalities beyond chest X-ray. In contrast to other studies, we intentionally do not train from scratch and use small datasets to explore the feasibility of diffusion in low-data and low-compute environments.

## 3. Methods

### 3.1. Image generation

All images are generated with Stable Diffusion v2.0, using an interactive open-source web interface (Rombach et al., 2022; AUTOMATIC1111, 2022). Images are sampled using the ancestral Euler scheduler (Karras et al., 2022). The main inference parameters influencing image generation quality are the number of steps for the sampling scheduler and the classifier-free guidance (CFG) scale (Ho and Salimans, 2022). Using more steps for sampling typically leads to better image quality but increases the inference time. The CFG scale can be used to set the trade-off between sample quality and sample diversity. A high CFG scale makes the model follow the text prompt more closely at the expense of diversity. Conversely, a low CFG scale results in images that deviate more from the prompt and consequently have lower fidelity but higher diversity.

To introduce a medical modality as a new concept to a pre-trained diffusion model, we use Textual Inversion (Gal et al., 2022). This process finds a vector in the text embedding space which optimally represents the concept. Practically, this is done by freezing the entire architecture apart from the embedding vector and performing backpropagation with a similarity loss, as illustrated in Figure 1. We train embeddings with a constant learning rate of 0.005 for 50,000 steps with a batch size of 1, which takes approximately 4 hours on an RTX2070 GPU. In the work of Gal et al., prompts are generated during training from a list of templates, for instance: "*a photo of a <embedding>*" or "*a rendering of a*

$<embedding>$". Since this does not necessarily apply to a medical imaging context, we prompt the model only with "$<embedding>$" during training.

We experiment with the number of sampling steps, the CFG scale, the number of images used to train embeddings, and the embedding vector size. To evaluate the impact of these parameters on generation quality, we compute the Fréchet Inception Distance using 1000 generated samples compared to 1000 real examples for each parameter setting (Szegedy et al., 2015). FID scores are calculated with an ImageNet pre-trained networks (FID), and a domain-specific medically pretrained network, RadImageNet (MFID) (Mei et al., 2022).

To explore the potential benefits of a diffusion-based approach over a GAN-based approach, we include the state-of-the-art StyleGAN3 as a baseline (Karras et al., 2021). To allow a fair comparison, we fine-tune a pre-trained StyleGAN3 on the same hardware for the same number of steps. A blind comparison between Stable Diffusion and StyleGAN3 was made by an expert prostate radiologist, who compared 50 pairs of images generated by the two methods, shown side-to-side and randomized. The radiologist indicated his preference for each of the 50 pairs and wrote down general impressions on the generation quality.

To investigate the usability of the trained embeddings, we also experiment with combining multiple trained embeddings using composable diffusion (Liu et al., 2023). This method allows prompting with a combination of embeddings using an AND operator in the prompt, e.g., "$<cardiomegaly>$ AND $<pleural\ effusion>$" to generate an image with both cardiomegaly and pleural effusion present. Additionally, this method allows a weight to be given to each embedding to tune the strength of each embedding separately. In this study, we use this to experiment with interpolating between healthy and diseased states and to generate images with multiple diseases present.

## 3.2. Classification

For classification experiments, we train ResNet-18 models, pre-trained on ImageNet (He et al., 2016; Deng et al., 2009). Models are trained with a fixed learning rate of $10^{-4}$ with the Adam optimizer for 6250 batches of 32 images on various combinations of real and synthetic data (Kingma and Ba, 2017). AUC is evaluated on the validation set during training, and performance of the best validation checkpoint on the test set is reported. We apply random horizontal flipping, gaussian noise, intensity transformations, channel dropout, translation, scaling and rotation as data augmentation.

## 3.3. Datasets

### 3.3.1. MULTI-MODAL MRI - PI-CAI

The main dataset used in this work is a recently released public dataset of 1500 prostate MRI cases. This dataset was released as part of the PI-CAI (Prostate Imaging: Cancer AI) challenge, where the task is to detect clinically significant prostate cancer (Saha et al., 2022). Each case is a 3D MRI scan featuring three modalities: T2-weighted imaging (T2W), apparent diffusion coefficient maps (ADC), and diffusion-weighted imaging (DWI). Since this work adapts a pre-trained 2D diffusion model, we extract one 2D axial slice per case. Each case is first resampled to a resolution of $3 \times 0.5 \times 0.5$ mm and then center-cropped to a $90 \times 150 \times 150$ mm ($30 \times 300 \times 300$ px) region. We select the median prostate slice for negative cases using the provided full prostate segmentations. We select the slice with

the maximum tumor area for positive cases according to the provided tumor segmentation maps. Each slice is finally upsampled to $512 \times 512$ px. Each modality is encoded as one of the RGB channels when training multi-modal embeddings. The training, validation and test set of the classification experiments each consist of 100 randomly sampled negative slices and 100 randomly sampled positive slices. The embeddings are trained on the training set.

### 3.3.2. Chest X-ray - CheXpert

CheXpert is a large public dataset of 224,316 chest radiographs, with corresponding labels for 14 different observations (Irvin et al., 2019). Since we explicitly investigate compositional prompting with the learned embeddings, we only sample images with a single class present. Specifically, we sample 100 AP-view radiographs to train embeddings for the following four observations: No Finding (healthy), Cardiomegaly, Pleural Effusion and Pneumonia. Each radiograph is first cropped to non-zero borders. Then, the longest edge is resized to 512 px, while keeping the aspect ratio fixed. Finally, the image is zero-padded to a square resolution of $512 \times 512$ px. The training, validation, and test set for the classification experiments each consist of 100 healthy and 100 cardiomegaly samples.

### 3.3.3. Histopathology - PatchCamelyon

PatchCamelyon is a public dataset of 327,680 $96 \times 96$ px patches extracted from histopathology whole-slide images of lymph node sections, originally released as part of the Camelyon16 challenge (Veeling et al., 2018; Ehteshami Bejnordi et al., 2017). Each patch has a corresponding binary label indicating the presence of metastatic tissue. We randomly select 100 negative and 100 positive patches for the training set. We use the official validation and testing splits of 32,768 cases each. All images are upsampled to $512 \times 512$ px.

## 4. Experiments

### 4.1. Adapting TI parameters to medical imaging

All embeddings in this section were trained using 2D T2-weighted healthy prostate slices. T2-weighted images clearly show the anatomy and are, therefore, easiest to judge qualitatively. Table 1 shows the FID and MFID scores after varying the number of sampling steps, CFG scale, embedding size, and the number of training cases relative to our final configuration used in the remainder of the paper: embedding size of 64 vectors per token, 100 cases per class, 100 sampling steps and a CFG scale of 2.

In general, we find that the FID and MFID scores identify general trends, but that they are not optimal metrics to judge generation quality and have sizable error margins (see Appendix A). For this reason, optimal parameters were chosen by inspecting generation results visually as well. Figure 5 in Appendix B shows the effect of the parameters studied in this section visually on a single random seed. For reference, Appendix C shows that directly generating images without applying textual inversion, by prompting the pre-trained model to generate prostate MRI scans, results in highly unrealistic images.

| Steps | FID | MFID | CFG scale | FID | MFID | Embedding size | FID | MFID | Training cases | FID | MFID |
|---|---|---|---|---|---|---|---|---|---|---|---|
| 25 | 118 | 4.04 | 1 | **85** | 4.50 | 8 | 100 | 2.92 | 5 | 158 | **2.55** |
| 50 | 106 | 3.38 | 2 | 99 | **2.87** | 16 | 110 | 3.22 | 10 | 106 | 3.25 |
| 75 | 101 | 3.11 | 3 | 146 | 4.51 | 32 | 149 | **2.86** | 50 | **96** | 3.41 |
| 100 | **99** | **2.87** | 5 | 173 | 61.4 | 64 | **99** | 2.87 | 100 | 99 | 2.87 |

Table 1: FID ($\downarrow$) and MFID ($\downarrow$) scores for embeddings generated with a varying number of sampling steps, CFG scale, embedding size, and a number of training cases. All settings are varied against 100 steps, CFG scale 2, embedding size 64, and 100 training cases.

| #Real | #Synthetic | AUC - Prostate MRI | AUC - Cardiomegaly | AUC - Histopathology |
|---|---|---|---|---|
| 200 | 0 | $0.780 \pm 0.017$ | $0.732 \pm 0.021$ | $\mathbf{0.878 \pm 0.011}$ |
| 200 | 2000 | $\mathbf{0.803 \pm 0.009}$ | $\mathbf{0.737 \pm 0.019}$ | $0.862 \pm 0.017$ |
| 0 | 200 | $0.737 \pm 0.019$ | - | - |
| 0 | 2000 | $0.766 \pm 0.020$ | - | - |
| 200 | 200 | $0.773 \pm 0.015$ | - | - |
| 0 | 2000* | $0.562 \pm 0.036$ | - | - |
| 200 | 2000* | $0.745 \pm 0.012$ | - | - |

Table 2: Mean test AUC $\pm$ standard deviation over 10 training runs for binary prostate cancer, cardiomegaly, and histopathology classifiers. Synthetic cases marked with an asterisk (*) were generated with an embedding trained on only 10 cases instead of 100.

## 4.2. Comparison to StyleGAN3

Images generated by the fine-tuned StyleGAN3 model achieved an FID score of 53, and an MFID score of 0.12, substantially lower than those shown in Table 1. However, in the blinded head-to-head comparison, the expert radiologist preferred the images generated by Stable Diffusion (36/50 images, 72%). There were more anatomically incorrect images generated by StyleGAN3, and often, the images had low contrast or were very dark. Similar to Section 4.1, this indicates that FID is not a particularly informative metric for comparing two architectures in a medical setting. Sets of 16 randomly generated images by both Stable Diffusion and StyleGAN3 are included in Appendix D and E, respectively.

### 4.3. Classification with synthetic data

In this section, we experiment using synthetic data to train classification models on multi-modal prostate MRI, chest X-ray and histopathology images. Embeddings are trained on two sets of 100 cases, with only negative or only positive cases. With these embeddings, up to 1000 cases for each class are generated, and combinations of real and synthetic data are used to train classification models. Similar to before, we perform more extensive experiments with multi-modal prostate MRI. Results are shown in Table 2, showing that for prostate MRI augmenting the 200-case training set with 2000 synthesized cases leads to a 2% improvement in AUC, from 0.78 to 0.80. These 2000 synthesized cases are based on embeddings trained with the same 200-case set used to train the classification models. This shows that generated cases can add non-trivial variation to the data distribution and that the embedding does not simply reproduce training cases. Furthermore, models trained with

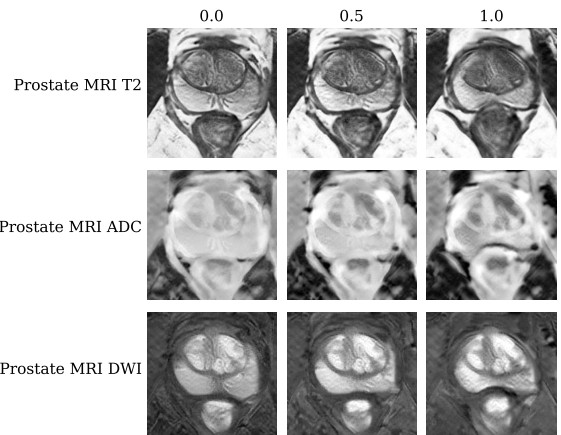

Figure 2: Interpolation between a healthy and diseased state for multi-modal Prostate MRI. The column titles show the trade-off between healthy and diseased.

only synthetic cases do not see a large drop in performance, indicating that the synthetic cases are diagnostically accurate. To confirm visual results from section 4.1, classification models trained with synthetic cases generated with embeddings trained on 10 cases instead of 100 show a dramatic drop in performance. This confirms that more cases are needed for Textual Inversion on medical data.

For cardiomegaly, however, including extra synthesized cases during training is hardly an added benefit. For histopathology, adding synthesized cases results in a performance drop of about 1%, which may indicate that synthetic cases are less useful for improving models that already attain high performance.

### 4.4. Composability of embeddings

In this section, we give preliminary evidence that composable diffusion works for medical data in two examples. In Figure 2, the disease state is gradually increased from healthy to diseased. The tumors in the prostate example become gradually more prominent (darker on ADC, brighter on DWI). Appendix F includes a more extensive figure. In Figure 3, multiple conditions are progressively added to a single healthy example. From a healthy image, pleural effusion, pneumonia, and cardiomegaly are added to the prompt for a single random seed. For the image with all three diseases, we gave each embedding a strength of 0.5 and found that increasing the CFG scale to 3 works better.

### 4.5. Controlling disease appearance with inpainting

This section demonstrates the potential of inpainting to control disease location precisely. Starting from a healthy example, a portion of the image is masked. The diffusion model denoises the masked part of the image, conditioned on a specific disease embedding. In Figure 4, the same healthy prostate example is masked in two locations with a different mask size. When inpainting conditioned on the positive embedding, this generates tumors at those locations of corresponding sizes. Similar to Section 4.4, this allows generating examples with specific disease appearance and could be useful for generating cases with rare tumor locations.

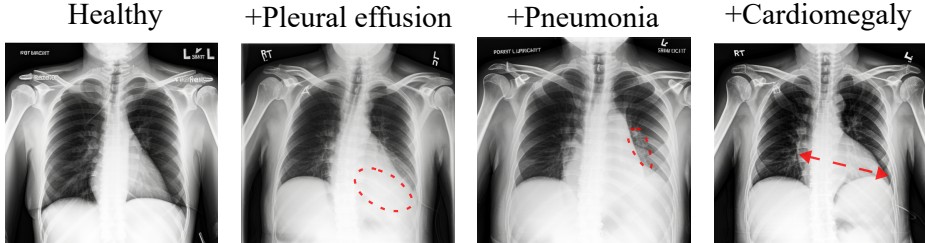

Figure 3: Visual example illustrating that multiple embeddings can be composed to show multiple pathologies in a single image. From left to right, pleural effusion, pneumonia, and cardiomegaly are progressively added to a healthy generated example.

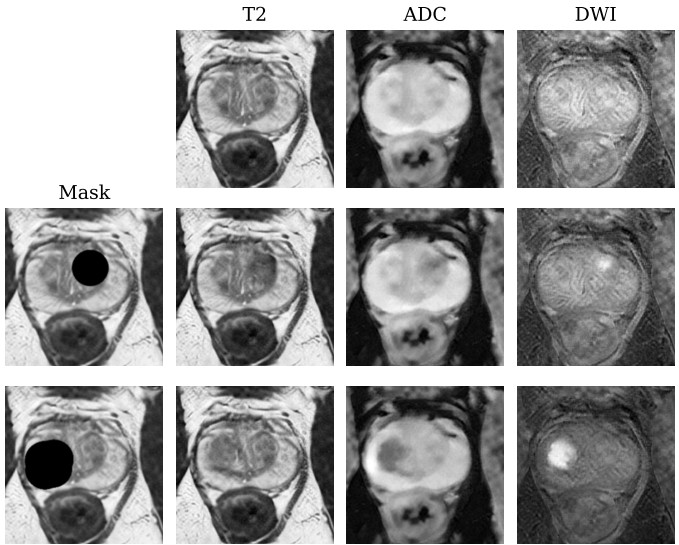

Figure 4: Inpainting of prostate cancer in different locations on the same healthy generated Prostate MRI example. The top row shows the original healthy case, with the bottom rows showing inpainting in different locations with varying mask sizes.

## 5. Conclusion

In this paper, we use Textual Inversion to demonstrate the adaptability of pre-trained latent diffusion models across various medical modalities. High-quality images can be generated using embeddings trained on 100 examples on a single consumer-grade GPU. Our showcased applications include enhancing diagnostic models in low-data scenarios by incorporating synthetic cases during training, simulating disease progression, and generating images with specific disease appearances. While a dedicated diffusion model trained on a large captioned medical dataset would likely yield superior results, our findings are promising for institutions with limited computational resources. This approach is particularly relevant for rare diseases where collecting large datasets is impractical. It remains viable and compatible with medically pre-trained models, including 3D models. Finally, the small file size of the trained embeddings may facilitate the sharing of medical information with reduced privacy concerns.

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

## Appendix A. FID reliability

To get an impression of the reliability of the FID metric, we estimated the 95% confidence interval for the MFID metric using bootstrapping. Calculation of the FID score based on 2048-lenght feature vectors is computationally expensive due to the linear algebra involved in computing the metric. For this reason, we estimated the 95% CI only for the MFID metric and our chosen configuration (100 steps, CFG 2, embedding size 64, 100 training cases) with $10^3$ repetitions, giving: 2.87 (1.34, 5.32).

This has significant overlap with most of the values in Table 1, so likely no hard conclusions can be drawn from that Table only. For a proper statistical comparison, a permutation test could be performed between distributions of FID scores calculated with random subsets of cases per setting. Since in this paper we used FID scores mostly to guide parameter choice, which we confirmed visually and with the classification experiments, we do not perform such rigorous (and expensive) comparisons.

## Appendix B. Textual Inversion parameters

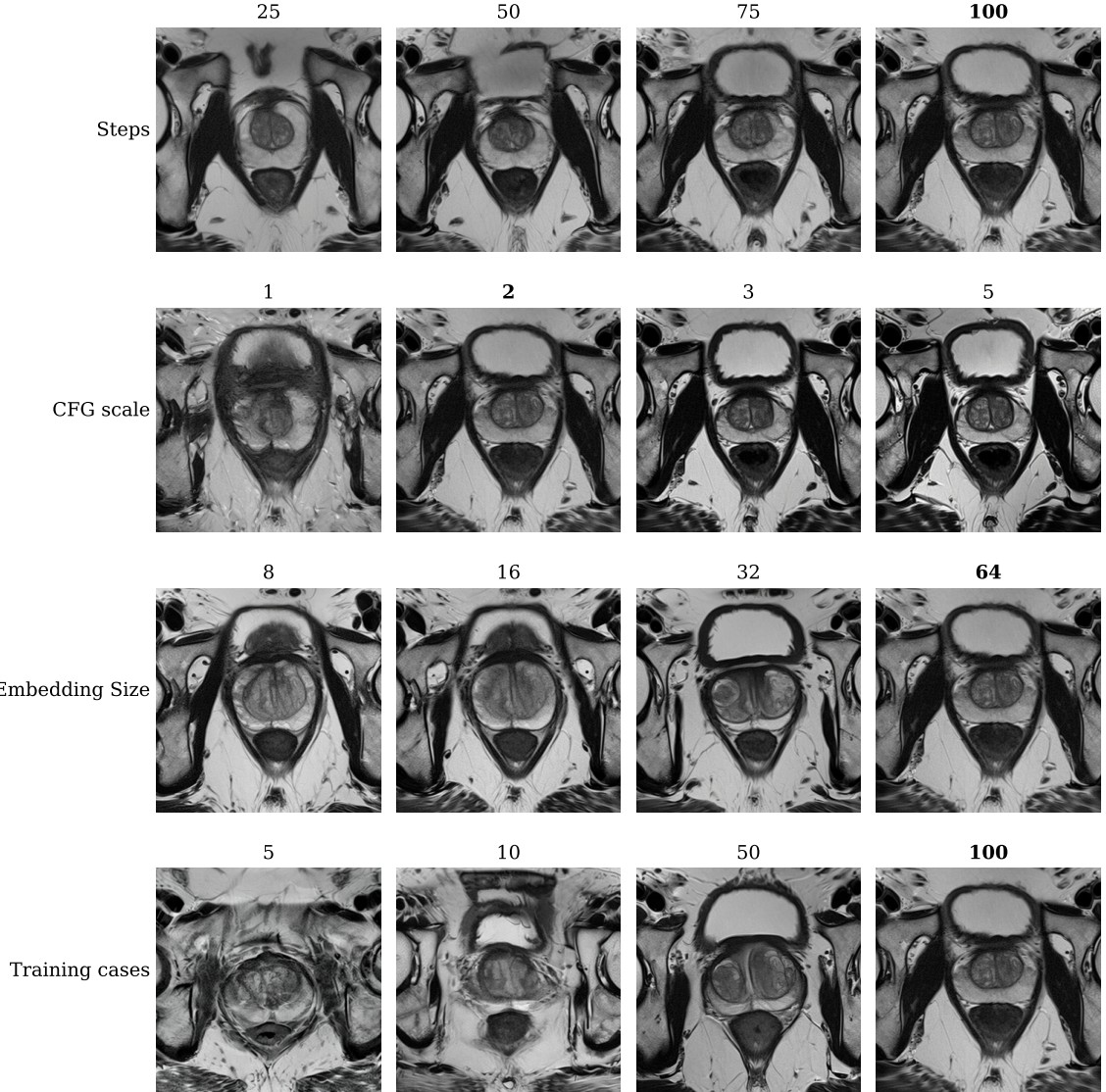

Figure 5: Visual examples illustrating the effect of varying inference and training settings for T2-weighted prostate MRI, all generated using the same random seed. Columns with a bold title indicate optimal values. Row labels indicate the parameter that changes along the column, with bold values set for the other parameters. For example, in the top row, the number of steps changes, but the CFG scale, embedding size, and training cases are 2, 64, and 100, respectively.

Figure 5 visually shows the effect of the parameters studied in Section 4.1 on a single random seed. A high number of sampling steps improves generation quality, with the generations for 25 and 50 steps showing incorrect anatomy for the bladder. Although a

CFG scale of 1 results in the lowest FID score in Table 1, visually, the results are much worse, featuring inaccurate general anatomy. A high CFG scale (e.g. 5 in Figure 5) also leads to bad results, showcased here by the simplified structure inside the prostate and a curious fractured pelvic bone. The difference between CFG scale 2 and 3 is not that large, but upon manual inspection, we find that a CFG scale of 2 gives better generations overall, as seems to be confirmed by the lower FID score in Table 1. The embedding size is optimally chosen to be large, with sizes 8 and 16 showing inaccurate generation, particularly of the bladder. Although size 32 looks better, the structure of the prostate itself is not nearly as good as generated by the size 64 embedding. Finally, the impact of the amount of training cases seems to trump all other settings, where 5 and 10 cases produce very unrealistic images. The embedding trained with 100 cases generates images with the most realistic prostate structure.

## Appendix C. Text-conditioned generation without textual inversion

This section demonstrates that a pre-trained Stable Diffusion model is not capable of generating MRI images of the prostate using text prompts. Chambon et al. (Chambon et al., 2022b) found that when prompting the model with "A photo of a lung xray", generated images look somewhat like real chest x-rays. For prostate images, we do not find the same. Figure 6 and 7 each show four random generations when prompting the model with "A prostate MRI scan" and "A T2-weighted MRI scan of a prostate", respectively. The output vaguely resembles medical scans, but is not close to a prostate MRI scan in any meaningful way. This demonstrates that for medical modalities that are less common, fine-tuning Stable Diffusion models is essential.

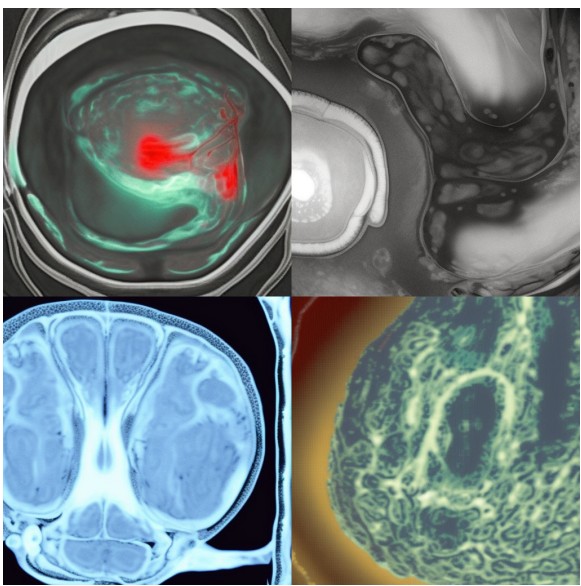

Figure 6: Four generated images when prompting a pre-trained Stable Diffusion model with "a prostate MRI scan"

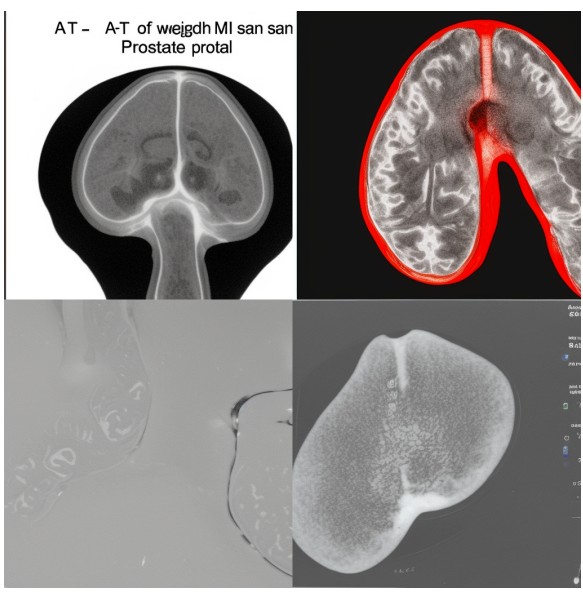

Figure 7: Four generated images when prompting a pre-trained Stable Diffusion model with "a T2-weigthed MRI scan of a prostate"

## Appendix D. Random sample of generated images - Stable Diffusion

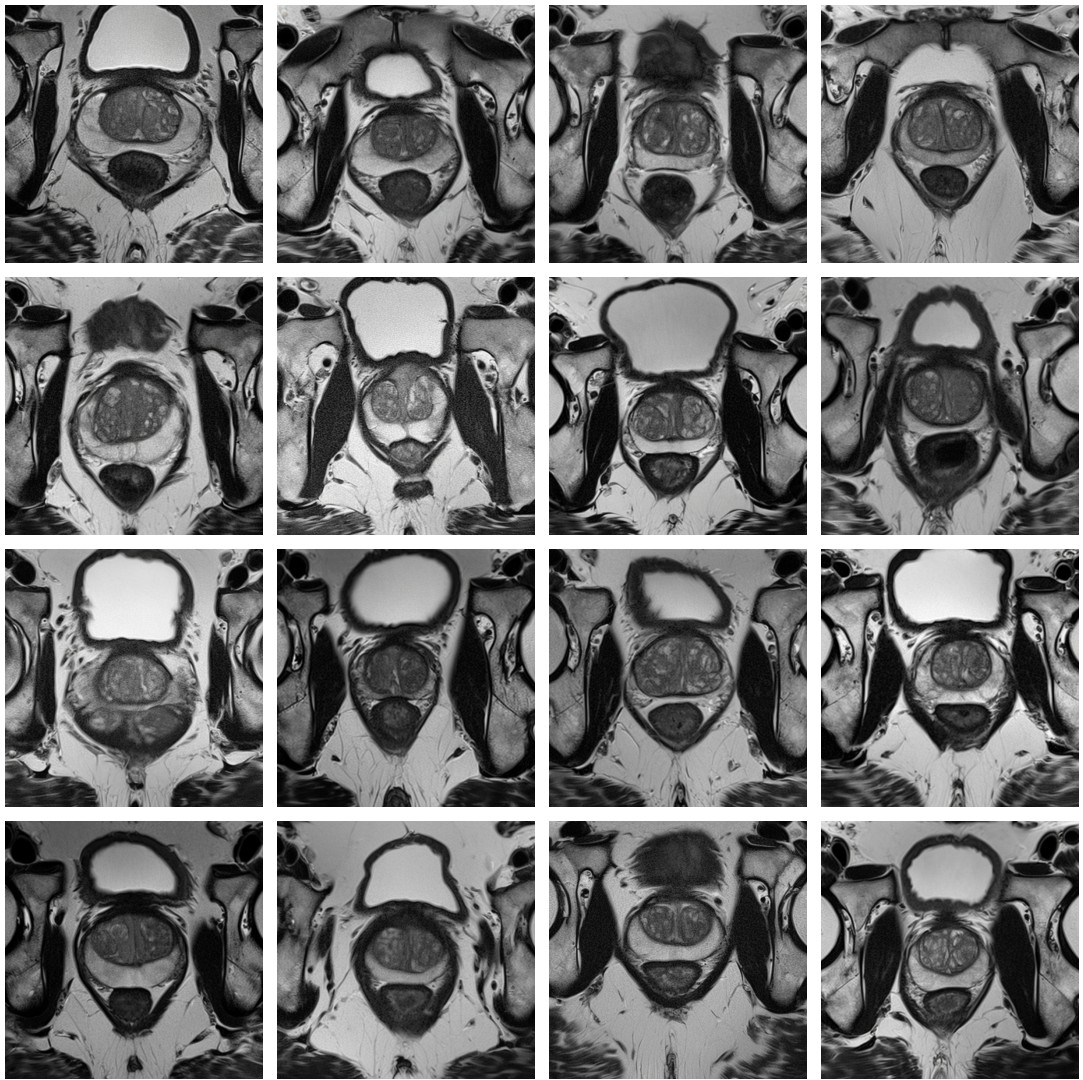

## Appendix E. Random sample of generated images - StyleGAN3

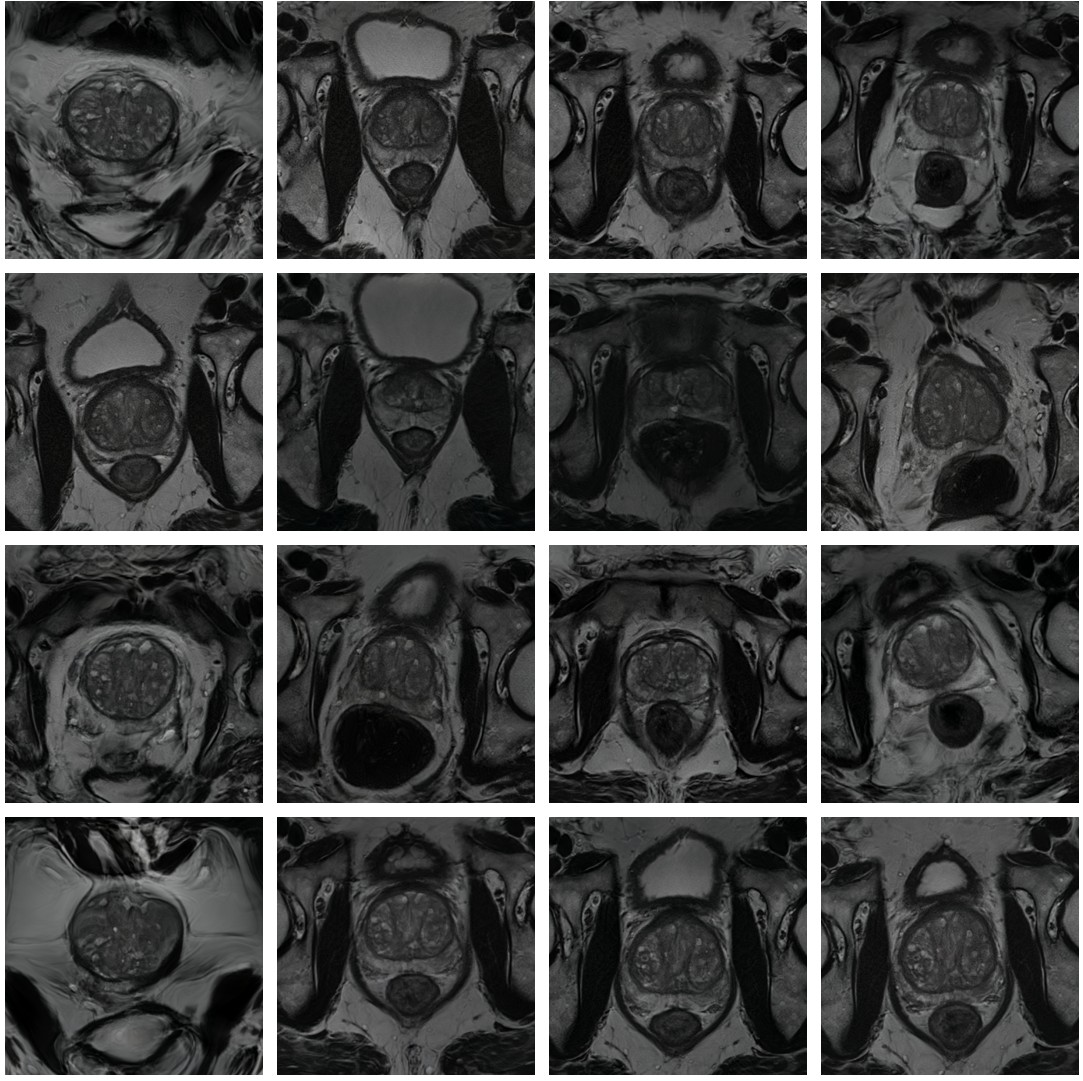

## Appendix F. Disease interpolation

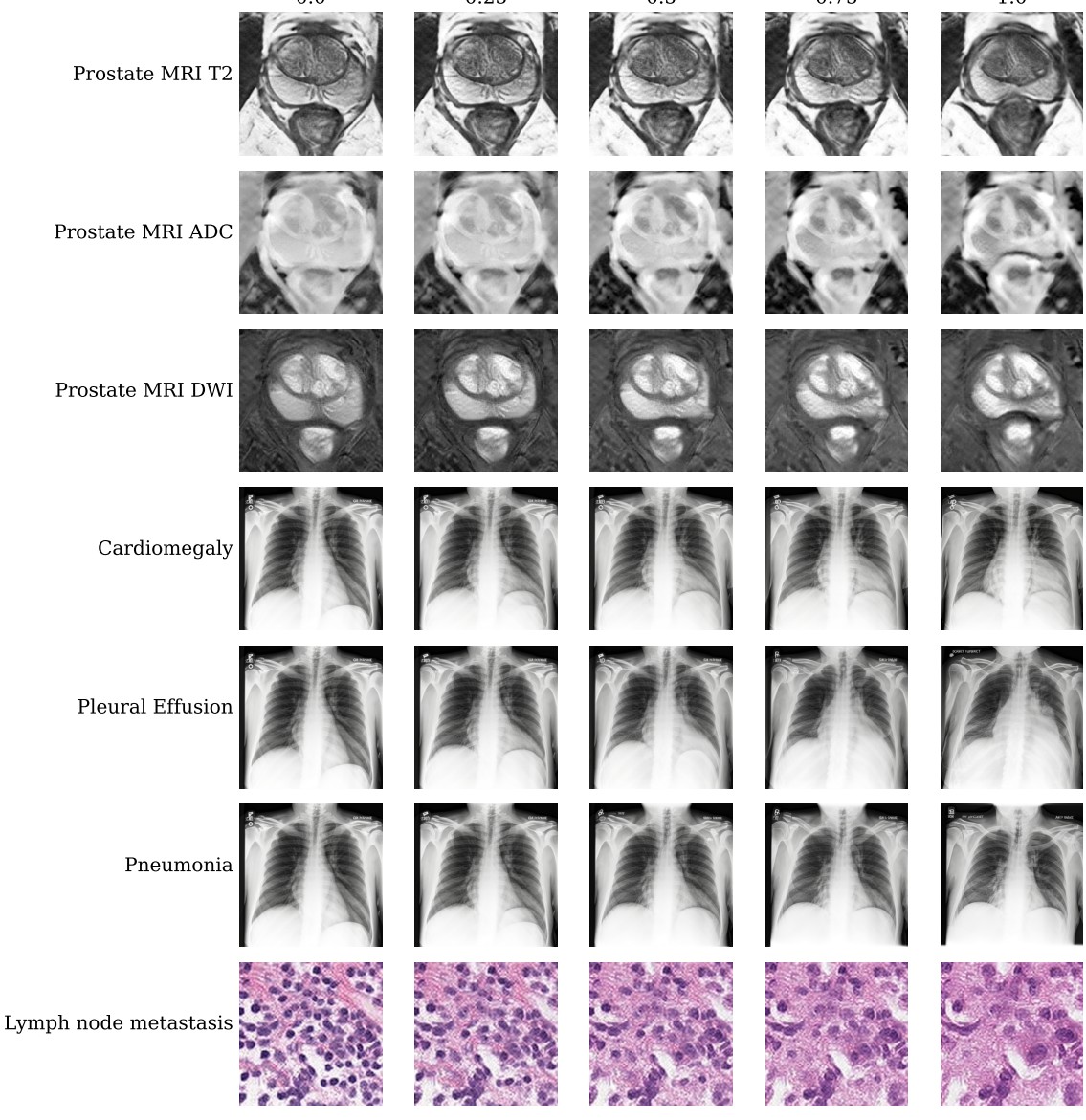

Figure 8: Visual examples illustrating interpolation between healthy and diseased states for multi-modal Prostate MRI, various pathologies on Chest X-ray, and lymph node metastasis in histopathology. The column titles show the trade-off between healthy and diseased. The Chest X-ray examples are all generated using the same random seed. The prostate images are cropped to the prostate region for visibility.

The disease state gradually increases from healthy to diseased, using composable diffusion. For instance, the cardiomegaly radiograph in the second column (25% diseased) is

generated with a prompt like "*0.25\*<healthy> AND 0.75\*<cardiomegaly>*". This seems to work well across the modalities studied in this paper: the tumors in the prostate example become gradually more prominent (darker on ADC, brighter on DWI); the heart in the cardiomegaly example appears to grow from left to right; the tissue in the lymph node metastasis example becomes gradually more abnormal.

