# OpenReview forum: "Medical diffusion on a budget: Textual Inversion for medical image generation"
_MIDL.io/2024/Conference — MIDL 2024 Oral_

### Official Review · Reviewer_ZT9S · 2024-02-27

**Confidence:** 5
**Preliminary Rating:** 4
**Final Rating:** 4

**Summary:**

In the text-to-image generation problem, the performance of existing works always suffers from limited availability of large pre-trained dataset. The authors show that pre-trained Stable Diffusion models can assist in medical image using textual inversion. The training is very light weighted and fast.

**Strengths:**

- Textual inversion is used in the proposed model to assist image generations.
- The diffusion model is fast to train and light weighted.
- Results show that the achieved images can assist downstream tasks and show good results for image classification.

**Weaknesses:**

- Is there any possibility to adopt the proposed method to a 3d model structure? Since the proposed method is tested on an MRI dataset which is originally a 3D dataset.
- The authors have implied in section 3.1 that the input sentence prompt shows like ‘a photo of  <embedding>’. Is there any possibility that a longer description of the image is passed, such as ‘a photo of a liver where an HCC is located at upper left, with size 3mm’. Would the model render a corresponding result? This could be more complicated than ”<cardiomegaly> AND <pleural effusion>”.
- What is the training time and training step of the diffusion model? Since the diffusion model always suffers from long training and inference time.
- Could you please show some other metric results, such as PSNR and SSIM?
- Does the sample size matter? Since the PatchCamelyon dataset is not originally 512\*512 and the authors upsampled them, while chest Xray is originally 512\*512.
- The authors seem to have missed some relevant literature. Specifically, they miss out on several relevant citations, e.g. “MedGen3D: A Deep Generative Framework for Paired 3D Image and Mask Generation”, and “Rethinking Semi-Supervised Medical Image Segmentation: A Variance-Reduction Perspective”. These relevant papers should be discussed.

**Detailed Comments:**

- Is there any possibility to adopt the proposed method to a 3d model structure? Since the proposed method is tested on an MRI dataset which is originally a 3D dataset.
- The authors have implied in section 3.1 that the input sentence prompt shows like ‘a photo of  <embedding>’. Is there any possibility that a longer description of the image is passed, such as ‘a photo of a liver where an HCC is located at upper left, with size 3mm’. Would the model render a corresponding result? This could be more complicated than ”<cardiomegaly> AND <pleural effusion>”.
- What is the training time and training step of the diffusion model? Since the diffusion model always suffers from long training and inference time.
- Could you please show some other metric results, such as PSNR and SSIM?
- Does the sample size matter? Since the PatchCamelyon dataset is not originally 512\*512 and the authors upsampled them, while chest Xray is originally 512\*512.
- The authors seem to have missed some relevant literature. Specifically, they miss out on several relevant citations, e.g. “MedGen3D: A Deep Generative Framework for Paired 3D Image and Mask Generation”, and “Rethinking Semi-Supervised Medical Image Segmentation: A Variance-Reduction Perspective”. These relevant papers should be discussed.

**Justification Of Final Rating:**

I would like to extend my thanks to the authors for their comprehensive classification work. The incorporation of further experiments and the broadening of classifications have thoroughly addressed all of my concerns.

====update===== Overall, I would like to vote for acceptance.

**Justification Of The Preliminary Rating:**

- Is there any possibility to adopt the proposed method to a 3d model structure? Since the proposed method is tested on an MRI dataset which is originally a 3D dataset.
- The authors have implied in section 3.1 that the input sentence prompt shows like ‘a photo of  <embedding>’. Is there any possibility that a longer description of the image is passed, such as ‘a photo of a liver where an HCC is located at upper left, with size 3mm’. Would the model render a corresponding result? This could be more complicated than ”<cardiomegaly> AND <pleural effusion>”.
- What is the training time and training step of the diffusion model? Since the diffusion model always suffers from long training and inference time.
- Could you please show some other metric results, such as PSNR and SSIM?
- Does the sample size matter? Since the PatchCamelyon dataset is not originally 512\*512 and the authors upsampled them, while chest Xray is originally 512\*512.
- The authors seem to have missed some relevant literature. Specifically, they miss out on several relevant citations, e.g. “MedGen3D: A Deep Generative Framework for Paired 3D Image and Mask Generation”, and “Rethinking Semi-Supervised Medical Image Segmentation: A Variance-Reduction Perspective”. These relevant papers should be discussed.

**Questions To Address In The Rebuttal:**

- Is there any possibility to adopt the proposed method to a 3d model structure? Since the proposed method is tested on an MRI dataset which is originally a 3D dataset.
- The authors have implied in section 3.1 that the input sentence prompt shows like ‘a photo of  <embedding>’. Is there any possibility that a longer description of the image is passed, such as ‘a photo of a liver where an HCC is located at upper left, with size 3mm’. Would the model render a corresponding result? This could be more complicated than ”<cardiomegaly> AND <pleural effusion>”.
- What is the training time and training step of the diffusion model? Since the diffusion model always suffers from long training and inference time.
- Could you please show some other metric results, such as PSNR and SSIM?
- Does the sample size matter? Since the PatchCamelyon dataset is not originally 512\*512 and the authors upsampled them, while chest Xray is originally 512\*512.
- The authors seem to have missed some relevant literature. Specifically, they miss out on several relevant citations, e.g. “MedGen3D: A Deep Generative Framework for Paired 3D Image and Mask Generation”, and “Rethinking Semi-Supervised Medical Image Segmentation: A Variance-Reduction Perspective”. These relevant papers should be discussed.

---

> ### Author Response · Authors · 2024-03-17
>
> We thank the reviewer for the thorough evaluation of our work, and the suggestions for improvement. Please see our detailed implementations and responses, point by point, below.
>
> > Is there any possibility to adopt the proposed method to a 3d model structure? Since the proposed method is tested on an MRI dataset which is originally a 3D dataset.
>
> In general, textual inversion can be applied to any text-to-image diffusion model, including 3D models. In this paper, we use Stable Diffusion, to show that even with a model pre-trained on natural images it is possible to fine-tune towards diagnostically relevant medical images. A requirement for this to work in 3D, would be a medically pre-trained 3D text-to-image latent diffusion model, which to the best of our knowledge is not yet openly available. Given such a model, we expect our approach to potentially learn faster and better embeddings, since the model has much more domain knowledge embedded. We have added a short line to the conclusion to attend the reader that our approach is compatible with 3D models.
>
> > The authors have implied in section 3.1 that the input sentence prompt shows like ‘a photo of <embedding>’. Is there any possibility that a longer description of the image is passed, such as ‘a photo of a liver where an HCC is located at upper left, with size 3mm’. Would the model render a corresponding result? This could be more complicated than ”<cardiomegaly> AND <pleural effusion>”.
>
> This is indeed possible and the authors of the Textual Inversion paper have experiments showcasing this possibility, but they noted that it gave worse results than simply prompting with the embedding name (Figure 10 in [1]). We did not do it in our paper, because it requires each sample in the training set to have an individual label describing the sample characteristics. For simplicity, we simply trained two embeddings per modality, one for the negative class and one for the positive class, prompting the model only with the embedding name, which we now clarified at the end of section 3.1, page 4. As a result, this approach can be applied to any dataset for which class labels are available, which is the most common scenario.
>
> Suppose for the PI-CAI data, we would have labels per scan describing location, appearance and size, the embedding could have been trained by prompting the model as you describe, e.g.:
>
> Scan 1: “a <prostate_mri_negative> with a benign lesion in the transitional zone”
>
> Scan 2: “a <prostate_mri_positive> of 5mm in the peripheral zone with irregular tumor boundary”
>
> Scan 3: ...
>
> Given our datasets and their lack of descriptive labels, we did not perform new experiments to test this approach.
>
> [1] Rinon Gal, Yuval Alaluf, Yuval Atzmon, Or Patashnik, Amit H. Bermano, Gal Chechik, and
> Daniel Cohen-Or. An Image is Worth One Word: Personalizing Text-to-Image Generation
> using Textual Inversion, August 2022
>
> > What is the training time and training step of the diffusion model? Since the diffusion model always suffers from long training and inference time.
>
> We train each embedding with a constant learning rate of 0.005 for 50,000 steps with a batch size of 1. This takes approximately 4 hours on an NVIDIA RTX2070. Since only the vector embedding is trained, few weight parameters are updated. Training the entire diffusion model from scratch indeed takes much longer, since all weights have to be optimized and the training datasets are typically orders of magnitude larger. Textual inversion leverages these big training efforts, by only optimizing a small set of weights on top of a pre-trained model. We have added this information to Section 3.1.

---

> ### Author Response · Authors · 2024-03-17
>
> > Could you please show some other metric results, such as PSNR and SSIM?
>
> PSNR and SSIM are metrics typically used to compare pairs images which should look exactly the same. In our case, since we have a generative model, generated images will never be identical to training or validation examples. In fact, this is a desirable property, since it allows the model to enrich small dataset and potentially improve model training as shown in Section 4.3.
>
> The best one can do is to compare distributions of real and generated images to each other to see if they are similar, which is why we employed the Fréchet Inception Distance, which is the standard metric for image synthesis. However, we realize that this metric is suboptimal and have therefore, also based on suggestion by another reviewer, expanded our validation approach:
>
> - FID is now calculated by comparing 1000 generated images to 1000 real images, instead of 100.
> - We estimate 95% confidence intervals with bootstrapping in Appendix A to give an impression of the reliability of the scores.
> - We introduce an extra metric, MFID, which is the FID score when using RadImageNet as a feature extractor, instead of an ImageNet pre-trained network.
>
> We hope that the combination of these extended metrics and the classification experiments convinces the reviewer that the generated images are realistic.
>
> > Does the sample size matter? Since the PatchCamelyon dataset is not originally 512*512 and the authors upsampled them, while chest Xray is originally 512*512.
>
> The main reason we upsampled the PatchCamelyon patches to 512*512 is that this was the resolution on which the Stable Diffusion v1.5 checkpoint was originally trained. In principle it is also possible to generate images of different resolution with the same model, but to ensure maximum model performance we kept most settings, including resolution, as close as possible to the pre-training scenario.
>
> > The authors seem to have missed some relevant literature. Specifically, they miss out on several relevant citations, e.g. “MedGen3D: A Deep Generative Framework for Paired 3D Image and Mask Generation”, and “Rethinking Semi-Supervised Medical Image Segmentation: A Variance-Reduction Perspective”. These relevant papers should be discussed.
>
> This field is moving very rapidly, and we thank the reviewer for pointing us towards some interesting recent work. We have included “MedGen3D: A Deep Generative Framework for Paired 3D Image and Mask Generation” in Section 2 discussing related work on generative imaging, as it discusses mask-conditioned imaging, which relates well to the inpainting scenario in Figure 4.
>
> The scope of our paper is mostly generative AI, specifically with diffusion models. Given the strict 8-page limit, we have chosen to not discuss “Rethinking Semi-Supervised Medical Image Segmentation: A Variance-Reduction Perspective”. This paper, as we understand it, is more within the scope of semi-supervised/transfer learning, which relates to our classification experiment and mentioned application of learning with few images/labels present. However, there is no generative modelling in the paper, and discussing how our work relates to the state-of-the-art semi-supervised work in the medical field would require a sizable extra paragraph with many relevant citations, for which there is unfortunately no space. We hope the reviewer understands our choice.

---

### Official Review · Reviewer_aBEv · 2024-02-27

**Confidence:** 4
**Preliminary Rating:** 4
**Final Rating:** 5

**Summary:**

The authors explore the application of textual inversion to a pre-trained diffusion model for natural images (stable diffusion) to enable the model to produce samples of different medical datasets. They explore using small numbers of samples for each dataset to perform textual inversion. The authors show these adapted models can be used to generate realistic-looking data, and samples generated from the model help to improve the accuracy of trained classifiers. The authors also show preliminary results with inpainting and composing embeddings to produce images with multiple pathologies.

**Strengths:**

The work is well explained and written. It presents a range of interesting experiments showing the promise of textual inversion. The authors test on three separate datasets and the classification results in particular are interesting.

I think it is useful to explore approaches that don't require large computational resources and datasets for training.

Code is made available.

**Weaknesses:**

The work is quite a light exploration of a few different applications, with no one area explored in any great detail.

It seems the authors tested their classification models on 200 images per dataset, but there is much more data available in these datasets. Why not test on more data?

**Detailed Comments:**

I didn't get a sense of whether stable diffusion could be used to produce medical images before the application of textual inversion. I think it would be helpful to do some text prompting of the model to try to generate the classes and see how those images look - I suspect given the large dataset it was trained on there will have been some images present and it may do an OK job.

I'm not sure it makes sense to calculate the FID with just 100 samples. It is typically recommended to calculate on 10,000 samples or more. This may be why the authors find the FID metric to not correlate well with observed sample quality. Perhaps the authors could estimate error bars, using bootstrapping, to understand if these measures are at all informative when calculated on small datasets.

Related, it might be best to use a network trained on medical images, rather than the Inception network, as the backbone when calculating the FID. The extracted features will likely be more relevant.

**Justification Of Final Rating:**

The authors have done a good job responding to my comments. In particular, I appreciate the new results investigating stable diffusion's native ability to generate prostate MRI, and the willingness to better explore the FID results with bootstrapping and using medical backbones. I'm upgrading my rating.

**Justification Of The Preliminary Rating:**

While a little 'lightweight' in some ways, I think the work is an interesting exploration of a few potential applications for textual inversion/ diffusion modelling and the community will find it interesting. Focus on methods that don't require lots of GPUs is valuable in the medical community.

**Questions To Address In The Rebuttal:**

How does stable diffusion do without any application of textual inversion?

Improvement of FID scores, either with more samples / a medical specific network.

---

> ### Author Response · Authors · 2024-03-17
>
> We thank the reviewer for the thorough evaluation of our work, and the suggestions for improvement. Please see our detailed implementations and responses, point by point, below.
>
> > The work is quite a light exploration of a few different applications, with no one area explored in any great detail.
>
> We have indeed chosen to focus on the wide applicability of textual inversion, and fine-tuning diffusion models in general, to showcase the potential applications in the medical field. We hope to provide enough evidence, for instance with the classification experiments, to convince readers of the validity and usefulness of our approach.
>
> > It seems the authors tested their classification models on 200 images per dataset, but there is much more data available in these datasets. Why not test on more data?
>
> We thank the reviewer for highlighting this potential improvement in the validation design. Our initial experiments were performed on the PI-CAI (prostate) dataset, which contains 1500 images of which 425 are cancer positive. For simplicity, and to avoid class imbalance, we sampled 50/50 negative/positive sets from this set. Given the total amount of 425 positive cases, we chose 100 samples for each class, for the train, validation and test sets. Later, we extended the classification experiments to CheXpert (cardiomegaly) and PatchCamelyon (histopathology) and kept the same sampling scheme.
>
> For CheXpert, out of the total 224,316 scans, there are only 671 which feature only cardiomegaly and no other disease, which is of the same order as the PI-CAI dataset.
>
> For PatchCamelyon, there are indeed much more patches available than we used in the paper. We have therefore re-run the experiments, using the same 100/100 negative/positive set for traing, but using the full official validation and test sets of 32,768 cases each (50/50 negative/positive).
>
> > I didn't get a sense of whether stable diffusion could be used to produce medical images before the application of textual inversion. I think it would be helpful to do some text prompting of the model to try to generate the classes and see how those images look - I suspect given the large dataset it was trained on there will have been some images present and it may do an OK job.
>
> This is indeed a good point to clarify in the paper. The first paper by Chambon we cite, does this for chest X-ray, and they found that Stable Diffusion models were able to produce images that resemble realistic chest X-ray scans when prompting with “a photo of a lung xray” [1]. We did the same and found that for prostate MRI this is not the case, with images generated by the diffusion model not being realistic. We have included example generations with prompts “a prostate MRI scan” and “a T2-weigthed MRI scan of a prostate” in Appendix C.
>
> [1] Pierre Chambon, Christian Bluethgen, Curtis P. Langlotz, and Akshay Chaudhari. Adapt-
> ing Pretrained Vision-Language Foundational Models to Medical Imaging Domains, Oc-
> tober 2022
>
> > I'm not sure it makes sense to calculate the FID with just 100 samples. It is typically recommended to calculate on 10,000 samples or more. This may be why the authors find the FID metric to not correlate well with observed sample quality. Perhaps the authors could estimate error bars, using bootstrapping, to understand if these measures are at all informative when calculated on small datasets.
>
> We have taken this suggestion to heart, and have extended the amount of samples for the FID calculation to 1000. We have not extended to the suggested 10,000, since the amount of real healthy prostate MRI scans in the PI-CAI is around 1000, and to avoid excessive generation time for the various training and inference settings in Table 1.
>
> Also, we have included a bootstrapping estimate in Appendix A to indicate the reliability of the FID scores. Indeed, we find that the error margins have substantial overlap with values in Table 1. Since we do not use only the FID metrics to confirm our parameter choice, but also visual inspection and classification experiments, we did not opt for a rigorous statistical comparison between parameter settings.
>
> > Related, it might be best to use a network trained on medical images, rather than the Inception network, as the backbone when calculating the FID. The extracted features will likely be more relevant.
>
> To explore the difference between an ImageNet pre-trained network and a medical pre-trained network, we have included FID scores based on RadImageNet [2]. Table 1 now features extra columns with the MFID scores, which indicate the FID scores based on RadImageNet.
>
> [2] Xueyan Mei, Zelong Liu, Philip M Robson, Brett Marinelli, Mingqian Huang, Amish Doshi,
> Adam Jacobi, Chendi Cao, Katherine E Link, Thomas Yang, et al. Radimagenet: an
> open radiologic deep learning research dataset for effective transfer learning. Radiology:
> Artificial Intelligence, 4(5):e210315, 2022

---

### Official Review · Reviewer_Gy3H · 2024-02-29

**Confidence:** 5
**Preliminary Rating:** 5
**Recommendation:** Oral
**Final Rating:** 5

**Summary:**

This paper explores using pre-trained latent diffusion models across various medical modalities through Textual Inversion; this process finds a vector in the text embedding space that represents the concept. Also, explored the usage of synthetic data to improve the performance of medical classification tasks, especially with limited computational sources. The authors have validated the models on MRI, ChestX-ray, and Histopathology datasets.

**Strengths:**

In their paper, the authors delve into the subject of enhancing model performance with limited data. They achieve this by utilizing diffusion models and training text embeddings using textual Inversion. This technique allows for the creation of more effective and accurate models, even when working with a smaller dataset. The authors tested the model on various medical modalities and achieved promising results.

**Weaknesses:**

I would like to express my appreciation for the well-written paper that provided a clear and concise explanation of the concepts. I am pleased to inform you that, based on my thorough review, no revisions or modifications are required.

**Detailed Comments:**

NA

**Justification Of Final Rating:**

he paper by the authors presents an approach to enhance model performance with limited data. By utilizing diffusion models and training text embeddings using textual inversion, the authors have created a technique that results in more effective and accurate models, even with smaller datasets. I would suggest that you accept the paper without making any changes to my previous decision.

**Justification Of The Preliminary Rating:**

The paper by the authors presents an approach to enhance model performance with limited data. By utilizing diffusion models and training text embeddings using textual inversion, the authors have created a technique that results in more effective and accurate models, even with smaller datasets. I would suggest to accept the paper.

**Questions To Address In The Rebuttal:**

NA

**Special Issue:**

No

---

> ### Author Response · Authors · 2024-03-17
>
> We are glad the reviewer appreciated the paper, and that there are no big points to address. Based on the comments of the other two reviewers we have made several additions and improvements:
>
> - The validation and test sets for PatchCamelyon are increased from 100 cases to the official splits consisting of 32,768 cases each.
> - The FID metrics were improved by increasing the number of cases for the generated and real sets from 100 to 1000. Additionally, we added Appendix A with a note on its reliability.
> - We added an extra FID metric, MFID, which is the FID score using RadImageNet as a feature extractor, instead of an ImageNet pre-trained network.
> - We have included prostate MRI generations by Stable Diffusion, without applying textual inversion, in Appendix C.

---

### Meta-Review · Area_Chair_fcBf · 2024-03-29

**Recommendation:** Accept (Poster)
**Confidence:** 4

**Metareview:**

The main strength of this paper lies in the extensive experiments, which demonstrates the efficacy of fine-tuning diffusion models on low computational budget.

However, the methodological contributions are rather limited, since this paper takes an existing model and simply demonstrates it in the case of different fine-tuning scenarios. I also emphasise that this paper clearly lacks a section to describe the employed latent diffusion model. This greatly reduces interest for non-experts in latent diffusion models, makes ablations difficult to parse, and decreases insights in the comparison with StyleGAN3.

Overall, I believe this work will lead to interesting discussions and I recommend acceptance. Finally, this paper is borderline for oral: despite the good scores, only 1 reviewer recommended oral, which is due to the paper being methodologically a bit light. I am personally leaning towards poster, but I'll leave the final decision to the program chairs.

---

### Decision · Program_Chairs · 2024-04-06

Accept (Oral)